# Xenon’s Sedative Effect Is Mediated by Interaction with the Cyclic Nucleotide-Binding Domain (CNBD) of HCN2 Channels Expressed by Thalamocortical Neurons of the Ventrobasal Nucleus in Mice

**DOI:** 10.3390/ijms24108613

**Published:** 2023-05-11

**Authors:** Nour El Dine Kassab, Verena Mehlfeld, Jennifer Kass, Martin Biel, Gerhard Schneider, Gerhard Rammes

**Affiliations:** 1Department of Anesthesiology and Intensive Care Medicine, Klinikum Rechts der Isar, School of Medicine, Technical University of Munich, 81675 Munich, Germany; nour.kassab@tum.de (N.E.D.K.); g.schneider@tum.de (G.S.); 2Department of Pharmacy-Center for Drug Research, Ludwig-Maximilians-Universitñt Mnchen, 81377 Munich, Germany; verena.mehlfeld@cup.uni-muenchen.de (V.M.); kass@vakzine-manager.de (J.K.); mbiel@cup.uni-muenchen.de (M.B.)

**Keywords:** xenon, anesthesia, electrophysiology, cyclic nucleotide-binding domain (CNBD), hyperpolarization-activated cyclic nucleotide-gated type-2 (HCN2), ventrobasal thalamus (VB), knocked-in hyperpolarization-activated cyclic nucleotide-gated type-2 mouse model, cyclic adenosine monophosphate (cAMP)

## Abstract

Previous studies have shown that xenon reduces hyperpolarization-activated cyclic nucleotide-gated channels type-2 (HCN2) channel-mediated current (I_*h*_) amplitude and shifts the half-maximal activation voltage ($V_{1/2}$) in thalamocortical circuits of acute brain slices to more hyperpolarized potentials. HCN2 channels are dually gated by the membrane voltage and via cyclic nucleotides binding to the cyclic nucleotide-binding domain (CNBD) on the channel. In this study, we hypothesize that xenon interferes with the HCN2 CNBD to mediate its effect. Using the transgenic mice model HCN2EA, in which the binding of cAMP to HCN2 was abolished by two amino acid mutations (R591E, T592A), we performed ex-vivo patch-clamp recordings and in-vivo open-field test to prove this hypothesis. Our data showed that xenon (1.9 mM) application to brain slices shifts the $V_{1/2}$ of I_*h*_ to more hyperpolarized potentials in wild-type thalamocortical neurons (TC) ($V_{1/2}$: −97.09 [−99.56–−95.04] mV compared to control −85.67 [−94.47–−82.10] mV; *p* = 0.0005). These effects were abolished in HCN2EA neurons (TC), whereby the $V_{1/2}$ reached only −92.56 [−93.16– −89.68] mV with xenon compared to −90.03 [−98.99–−84.59] mV in the control (*p* = 0.84). After application of a xenon mixture (70% xenon, 30% O_2_), wild-type mice activity in the open-field test decreased to 5 [2–10] while in HCN2EA mice it remained at 30 [15–42]%, (*p* = 0.0006). In conclusion, we show that xenon impairs HCN2 channel function by interfering with the HCN2 CNBD site and provide in-vivo evidence that this mechanism contributes to xenon-mediated hypnotic properties.

## 1. Introduction

The gaseous anesthetic xenon is a promising alternative to traditional anesthetics [1] in high-risk surgeries due to its effective anesthetic properties associated with very few side effects such as the absence of cardiovascular complications [2], low blood solubility [3], and protection against post-operative delirium [4]. Preclinical research has revealed that xenon exhibits neuroprotective properties including the reduction of Aβ toxicity, a hallmark of Alzheimer’s disease [5,6] and the reduction of neuronal cell loss after traumatic brain injuries [7].

In the central nervous system, xenon has a unique mechanism of action, acting as a low potent antagonist of α-amino-3-hydroxy-5-methyl-4-isoxazole propionic acid (AMPA) and N-methyl-D-aspartate (NMDA) receptors [5,8] and as an agonist of the two-pore-domain potassium channel (TREK-1) [9]. Unlike other inhalational anesthetics, xenon does not interact with $GABA_{a}$ receptors [10] and the TWIK-related acid-sensitive K+ (TASK) channels [9]. Previous studies have shown that xenon disrupts thalamocortical signal propagation by inhibiting the hyperpolarization-activated cyclic nucleotide-gated channel type 2 (HCN2) in thalamocortical neurons of the ventrobasal (VB) thalamic nucleus [11].

The thalamocortical network is responsible for feed-forward input into the cortex [12]. It has two main firing modes depending on the state of consciousness, short-bursts/high-frequency action potentials (AP) during sleep and tonic/single-spike AP in wakefulness [13]. In the tonic mode, thalamocortical neurons’ AP frequencies correlate with the strength of sensory inputs and are forwarded to the respective cortical region. In contrast, the burst mode is linked to a reduction of thalamic information going to the cortex [14].

HCN channels are implicated in the regulation of neurons’ resting membrane potential [15], synaptic transmission [15], dendritic integration [16], and pacemaker activity [17]. HCN channels conduct a potassium and sodium current known as (I_*h*_) and are activated upon membrane hyperpolarization to dampen responses to inhibitory or excitatory stimuli and stabilize the resting membrane potential [18]. HCN channels are expressed in four different isoforms (1–4) in mice and humans, all belonging to the superfamily of voltage-activated channels [10,19]. The channels are organized around the transmembrane core responsible for gating and ion conductance while the c-terminal region contains the highly conserved cyclic nucleotide-binding domain (CNBD), which binds cyclic nucleotides, and a C-linker peptide connecting the two domains [20]. The cytosolic N-terminal domain on the other hand varies considerably among the different HCN channel isoforms [21].

HCN2 channels are gated by the cell’s membrane potential and the neurotransmitter system via cyclic nucleotides cyclic adenosine monophosphate (cAMP) and cyclic guanosine monophosphate (cGMP) binding to the channels CNBD [15]. cAMP binding to HCN2 potentiates channel activation [22] and is also critical for switching thalamocortical neurons from burst to tonic firing during the transition from sleep to arousal [23]. The recently published knock-in mouse model HCN2EA, whereby the binding of cAMP to the HCN2 channel is abolished by the mutation of two amino acids (R591E, T592A), showed a functional HCN2 channel with an I_*h*_ and a half-maximal activation voltage ($V_{1/2}$) identical to wild-type, but with an I_*h*_ current insensitive to cAMP modulation [23].

Xenon impairs HCN2 function and thalamocortical signal propagation by reducing I_*h*_ current amplitude and causing a leftward shift to more hyperpolarized potentials of the $V_{1/2}$ [11]. However, after saturating the intracellular cAMP concentration, xenon failed to impair HCN2 channel function supporting the hypothesis that xenon interferes with cAMP binding to the CNBD of HCN2 channels [11].

In this study, we performed in vivo and ex vivo experiments to detail the pharmacological target of xenon on HCN2 channels and its relevance for xenon’s sedative effect in the HCN2EA model. The results indicate that a non-functional cAMP binding site in ventrobasal neurons (VB) affects the ability of xenon to inhibit HCN2 channels and plays a critical role in its hypnotic effect on the CNS, providing new insights into the molecular mechanisms underlying its sedative properties.

## 2. Results

### 2.1. Xenon Impairs HCN2 I_h_ Current of VB Neurons in Wild-Type Acute Brain Slice

In order to investigate the effect of xenon interacting with the CNBD of HCN2 channels on the electrophysiological properties of the channel, independent control recordings were first performed in a nitrogen gas mixture consisting of 65% N_2_, 30% O_2_, and 5% CO_2_ (Figure 1A schematic black rectangle, N_2_). The nitrogen gas mixture had the same composition as the xenon mixture, where nitrogen replaced xenon. As reported by Kratzer et al. (2017) [24], co-application of this nitrogen gas mixture with a carbogen gas mixture consisting of 95% O_2_, 5% CO_2_ (Figure 1A schematic brown rectangle, Carbogen) in the ACSF did not affect the cell viability and did not cause a change in resting membrane potential, action potential (AP) threshold, frequency, and HCN2 current (I_*h*_).

Our experiments confirmed these findings by comparing the I_*h*_ current after 30 min of nitrogen and carbogen co-application (Figure 1A wild-type: II, HCN2EA: IV) to carbogen alone (Figure 1A wild-type: I; HCN2EA: III). In wild-type and HCN2EA thalamocortical neurons, I_*h*_ traces were comparable when carbogen was applied alone, to when recorded with the nitrogen mixture in the artificial cerebrospinal fluid (aCSF) (Figure 1A wild-type: (I,II), HCN2EA: (III,IV)) and as well as the I_*h*_ activation curve (Figure 1B wild-type and Figure 1C: HCN2EA). The activation curve analysis in wild-type revealed no significant changes in the $V_{1/2}$ (Figure 1D, N_2_ (black hexagons): −85.67 [−88.22–−82.55] mV, *n* = 9; carbogen (brown hexagons): −85.52 [−88.22–−82.65] mV, *n* = 4; AUC = 0.5 [0.2–0.8]). This result was also true for recordings taken from HCN2EA thalamocortical neurons (Figure 1D $V_{1/2}$ N_2_ (black triangles): −88.52 [−94.42–−81.30] mV, *n* = 7) in the presence of N_2_ co-applied with carbogen compared to carbogen applied alone ((Figure 1D carbogen (brown triangles): −90.03 [−98.99–−84.59] mV, (*n* = 4); AUC = 0.4 [0.07–0.75]). A comparison of the $V_{1/2}$ between wild-type and HCN2EA in both conditions found no statistically significant differences between the values during N_2_+carbogen co-application or carbogen alone (*p* = 0.57; Kruskal–Wallis with post hoc false discovery rate correction (FDR), AUC = 0.7 [0.3–0.9]). Therefore, we can rule out any significant effect of gas co-application (N_2_+carbogen) on the electrophysiological properties of neurons from wild-type and HCN2EA animals. Hence, we reconfirmed the validity of the N_2_+carbogen co-application as a control [24].

Our experiment aimed to investigate the impact of xenon on the regulation of HCN2 channels by cAMP. We first performed whole-cell patch-clamp experiments on VB neurons under two different conditions using two different intracellular solutions, with cAMP (1 µM) and without cAMP (Figure 2A (I,II): 0 µM cAMP, (III,IV): +1 µM). A 1 µM cAMP was reported as a value close to the apparent cAMP affinities of I_*h*_ in neuron s [24], while 30 µM was used by Mattusch et al. (2015) to block xenon HCN2 inhibition, and represents a saturated concentration [11].

The addition of 1 µM cAMP resulted in an I_*h*_ with a slight increase in the I_*h*_ curve amplitude and kinetics typical for a cAMP-modulated I_*h*_ current (Figure 2A (I,III)). This resulted in a slight shift of the I_*h*_ activation curve toward a more depolarized potential (Figure 2B N_2_-wt: filled black hexagons, N_2_-wt+cAMP: open black hexagons) with $V_{1/2}$: −83.69 [−89.03–−76.72] mV, *n* = 5 compared to $V_{1/2}$: −85.67 [−94.47–−82.10] mV, *n* = 9 without cAMP (Figure 2C N_2_-wt: filled black hexagons, N_2_-wt+cAMP: open black hexagons). This difference was not statistically significant (*p* = 0.42; Kruskal–Wallis test, AUC = 0.4 [0.1–0.7]).

In wild-type slices, after a 30 min incubation with the xenon gas mixture, there was a decrease in I_*h*_ current amplitude (Figure 2A N_2_: I and III xenon: II and IV) and a significant shift in the voltage activation toward a more hyperpolarized membrane potential compared to the respective controls independent of the added cAMP (Figure 2B, xenon: filled and open green hexagons, N_2_: black hexagons). In the absence of 1 µM cAMP, the $V_{1/2}$ shifted significantly after xenon incubation in wild-type animals from −85.67 [−94.47–−82.10] mV (*n* = 9) to −97.09 [−99.56–−95.04] mV (*n* = 11), (Figure 2C; filled black (N_2_-wt) compared to filled green (Xe-wt) hexagons, *p* = 0.0009; Kruskal–Wallis test, AUC = 0.9 [0.8–1]). A similar effect could be observed when 1 µM cAMP was added to the intracellular solution. The $V_{1/2}$ of N_2_-wt+cAMP ($V_{1/2}$: −83.69 [−89.03–−76.72] mV, *n* = 5) shifted to more hyperpolarized potentials ($V_{1/2}$: −94.78 [−96.36–−88.44] mV; *n* = 7) upon xenon application, (Figure 2C, N_2_-wt+cAMP: open black hexagons compared to Xe-wt+cAMP: open green hexagons, *p* = 0.047; Kruskla–Wallis test, AUC = 0.8 [0.5–1]). No statistically significance was found between $V_{1/2}$ of xenon with and without 1 µM cAMP (Figure 2C, Xe-wt+cAMP: open green compared to Xe-wt: filled green hexagons, *p* = 0.1017; Kruskal-Wallis test, AUC = 0.2 [0–0.5]). The results indicated that xenon can inhibit thalamocortical neurons HCN2 channels in acute brain slices from adult wild-type mice and that 1 µM cAMP is not potent enough to interfere with xenon HCN2 channels inhibition.

### 2.2. Cyclic Nucleotide Binding Domain (CNBD) Mutation Reduces the Inhibitory Effect by Xenon on I_h_

To further elaborate on the mechanism by which xenon affects I_*h*_ currents via the CNBD site of HCN2 channels, we used a transgenic mice model HCN2EA in which the cAMP binding site was abolished. HCN2EA channels are generated by two amino acid substitutions in the CNBD (R591E and T592A) [23]. These residues are located in the loop between the P helix and the 7β sheet of the CNBD site [25,26] and directly interact with the cAMP phosphate group for high-affinity binding. Mutation to one or both residues is sufficient to abolish HCN regulation by cyclic nucleotides [27,28].

Xenon application for 30 min did not change the I_*h*_ curve shape (Figure 3A, traces I and III) or shift the activation curve recorded in HCN2EA neurons (Figure 3B). The HCN2EA $V_{1/2}$ of I_*h*_ during xenon incubation (Xe-EA) was −92.56 [−93.16–−89.68] mV *n* = 6 compared to the respective control N_2_-EA ($V_{1/2}$: −90.03 [−93.16–−89.68] mV, *n* = 7). The difference was not statistically significant (Figure 3C, *p* = 0.83; Mann–Whitney test, AUC = 0.54 [0.14, 0.85]).

Since the $V_{1/2}$ values of wild-type and EA control recording overlapped considerably, their calculated AUC value was lower than 0.7 (low effect size), and the confidence interval of their AUC included 0.5, indicating no significant effect due to the CNBD mutation on the $V_{1/2}$ of I_*h*_. For those reasons, we pooled all the control experiments in a single group termed control, achieving a sample size of *n* = 16. Finally, we compared the control group with wild-type_(*Xe*)_ and HCN2EA_(*Xe*)_ (Figure 3D,E, control: black hexagons, HCN2EA_(*Xe*)_: green triangles, and wild-type_(*Xe*)_: green hexagons). We found a significant difference in the $V_{1/2}$ between the control group ($V_{1/2}$: −86.94 [−95.85–−83.79] mV; *n* = 16), compared to the wild-type_(*Xe*)_ ($V_{1/2}$: −97.09 [−99.56–−95.04] mV; *n* = 11), (Figure 3E, control: black hexagons compared to wild-type_(*Xe*)_: green hexagons, *p* = 0.0012; Kruskal-Wallis test, AUC = 0.8 [0.6–1]). No significant difference was found between the $V_{1/2}$ of the control group ($V_{1/2}$: −86.94 [−95.85–−83.79] mV; *n* = 16) and that of HCN2EA_(*Xe*)_ ($V_{1/2}$: −92.56 [−93.16–−89.68] mV; *n* = 6), (Figure 3E, control: black hexagons compared to HCN2EA_(*Xe*)_: green triangles, *p* = 0.91; Kruskal-Wallis test, AUC = 0.6 [0.4–0.8]). In the presence of xenon, the $V_{\left(1/2\right)}$ recorded in wild-type (green hexagons) were significantly more hyperpolarized than the one recorded in HCN2EA (Figure 3D, HCN2EA_(*Xe*)_: green triangles compared to wild-type_(*Xe*)_: green hexagons, *p* = 0.01; Kruskal-Wallis test, AUC = 0.98 [0.9–1]). These results indicate that the xenon inhibition of the HCN2 channel, observed in the wild-type can be significantly reduced by a mutated HCN2 cyclic nucleotide-binding domain (CNBD).

### 2.3. Xenon Sedation Is Absent in The HCN2 Knock-In Mouse Model (HCN2EA)

In the next step, we investigated the contribution of the HCN2EA mutation to the sedative effects of xenon by conducting an in vivo experiment in a modified open-field test (Figure 4A timeline schematics, B open-field test schematics). We quantified the number of lines crossed during the habituation and exploration period and the general activity during xenon application. Wild-type and HCN2EA mice behaved similarly during the habituation (Figure 4A habituation, timelines from min −5 to 0) with a mean of 55 [42–65] lines crossed for wild-type (*n* = 5, grey circles) and 60 [55–62] lines crossed for HCN2EA animals (*n* = 5, grey triangles). The difference was not statistically significant (Figure 4C, habituation, $p_{\left(habituation\right)}$ = 0.7069; Mann–Whitney test; Appendix A of WT and HCN2EA mice).

During the exploratory behavior period (Figure 4A, from min 0 to 10), the control nitrogen gas (70%N_2_, 30%O_2_) was applied. Wild-type animals (*n* = 7, black circles; 45 [40–50] lines crossed) did not behave significantly different compared to their HCN2EA littermates (*n* = 6, black triangles; 53 [50–58] lines crossed), Figure 4C, $p_{\left(controlgas\right)}$ = 0.2128; Mann–Whitney test). These results indicated that neither the test environment nor the gas influx into the box affected the behavior of the genetically modified animals.

When the gas was then switched from a nitrogen mixture to a xenon mixture (Figure 4A, 70%xenon, 30%O_2_, from min 10 to 20), the average activity of the wild-type mice (*n* = 7, green circles) rapidly decreased from min 12 to 14 until it reached 5 [2–9]% between min 18 to 20 (Figure 4D, Appendix A). In contrast, HCN2EA mice (*n* = 7, green triangles) showed no significant decrease in average activity before min 16 after xenon application and maintained a level close to 30 [20–40]% in the last 2 min. The linear regression of the averaged activity of wild-type and HCN2EA mice was plotted and compared by two-way ANOVA. The results showed a significant difference between the two regression traces (*p* = 0.0006; two-way ANOVA) with the factor of xenon sedation contributing the most to the variation between them (20.38%). This suggests that the difference in activity levels between the two groups, wild-type, and HCN2EA, is primarily driven by changes in activity levels over time (period of xenon sedation). These in-vivo results demonstrate that the CNBD site of HCN2 channels plays a crucial role in mediating the xenon sedative effects.

## 3. Discussion

In this study, we demonstrate that the sedative effect of the gaseous anesthetic xenon is strongly dependent on the CNBD site of HCN2 channels. These findings provide additional insight into how xenon may disrupt thalamocortical signal propagation. The whole-cell patch-clamp results of wild-type thalamocortical neurons show a significant leftward shift in the $V_{1/2}$ of I_*h*_ after xenon application, which is consistent with the results seen by Mattush et al. (2015) [11]. On the other hand, this effect was significantly diminished in thalamocortical neurons of HCN2EA mice. This result supports the hypothesis that xenon interacts with the CNBD of HCN2 channels and is crucial for mediating HCN2 channel inhibition. Finally, the in vivo experiments showed that xenon interference with the CNBD site is necessary for mediating part of xenon’s sedative effect.

Xenon has been in clinical practice for more than 70 years [29] and is currently considered to be an ideal anesthetic due to its beneficial anesthetic properties and neuroprotective effects. Continuous studies have shown that xenon reduces excitotoxicity in different disease models, such as Aβ toxicity [5] and traumatic brain injury [30]. Continuous improvement of extraction techniques has already reduced the price to a comparable level with other inhaled anesthetics [31]. Xenon would be favored over other anesthetics in high-risk surgeries [4] and is currently in a clinical trial for use in pediatric surgery [32]. Consequently, unraveling the pharmacological mechanism of how xenon produces sedation and anesthesia helps to assess its clinical use regarding the application to patients’ risk and safety.

Previously, PET scan imaging has shown that xenon reduces the whole-brain metabolic rate and significantly decreases cerebral activity in the thalamus [33], which has a key role in the anesthesia-dependent loss of consciousness. Our investigation focused on thalamocortical relay neurons in the VB due to their crucial role in sensory information transfer to the cortex [34]. Both murine and human thalamic regions exhibit similar distributions of HCN channels in the VB [35,36]. The xenon mixture used in this study (65% xenon, 30% O_2_, 5% CO_2_) resulted in a final xenon concentration of 1.9mM, which is close to the human equivalence of 1 MAC [11,24]. This concentration does not represent however the full mouse MAC and is still regarded as a sub-anesthetic concentration. Therefore, we hold the assumption that the 1.9 mM concentration used in our experiments mediates a more pronounced effect on human neurons.

The thalamocortical neurons in the VB nucleus express high levels of HCN2 and HCN4 channels. They are expressed in different cellular compartments on thalamocortical neurons [23,37]. HCN4 channels are more enriched at the dendritic region of the neurons while HCN2 channels are more somatically enriched [37]. HCN2 was presumed to be the predominant subtype in the resting membrane potential regulation since knocking it out reduces the I_*h*_ current [38]. Recent studies with brain-specific deletion of HCN4 also showed a marked decrease in I_*h*_ in the VB neurons, indicating that both channels contribute to thalamocortical neurons resting membrane potential regulation [39]. However, cAMP-modulated I_*h*_ current in the mice VB is mainly driven by HCN2 channels [39] even though HCN4 can bind cAMP with high affinity, its expression in the VB is not sufficient enough to rescue the cAMP-modulated current in the I_*h*_ in the HCN2EA VB [23]. The entire currents were recorded from the somata of thalamocortical neurons thereby limiting the impact of dendritic HCN4 on total I_*h*_. In contrast, HCN4 might contribute to some extent to the xenon effects observed in the in vivo experiments.

HCN2 channels are partially open at rest [15] and are fully activated at hyperpolarized potentials where I_*h*_ produces a depolarizing inward current that limits the cell’s hyperpolarization and allows the timely re-initiation of AP [19]. This allows I_*h*_ to play the role of a pacemaker current and ensures rhythmic activity in the thalamus [17]. When cAMP binds to HCN2 at the CNBD site, it causes a conformational change that shifts the CNBD position away from the transmembrane domain and relieves the domain from the CNBD-dependent inhibition [40]. This mechanism upregulates HCN2 activity and shifts the I_*h*_ activation curve to a less-negative potential [10]. Thus, I_*h*_ activates more rapidly and completely already at early hyperpolarized potentials [19]. Xenon in contrast produces the opposite effect on I_*h*_ activation and amplitude demonstrated in a previous [11] and the present study.

Xenon shifts the HCN2 voltage activation curve to more negative potentials and decreases the I_*h*_ current amplitude, thus downregulating HCN2 activity. As a result, HCN2 channels will have a lower open probability and less impact on neuron activities. Additionally, cyclic nucleotides positively modulate HCN2 via the CNBD site and negatively via the c-terminal end by a cGMP-dependent protein kinase II [41]. The concerted synthesis of cAMP and cGMP produces an HCN2-dependent fine-tuning of the resting potential and pacemaker activity [41]. The xenon-induced interference with the cAMP binding site may distort this balance, finally resulting in channel inhibition. Our experiments with HCN2EA mouse single-cell recordings showed that abolishing the cAMP sensitivity of HCN2 channels results reduces significantly xenon’s ability to inhibit the HCN2 channel as well. Thereby, we conclude that xenon interferes with the CNBD site on HCN2 resulting in channel inhibition.

Moreover, cAMP-sensitive upregulation of HCN2 activity not only modulates the channel but is also essential for switching the thalamocortical-network firing mode from burst to tonic and subsequently allows the transfer of sensory information to the cortex [14,42]. This cAMP-dependent regulation provides a counterweight against inhibitory currents and locks neurons in transmission mode [43]. The abolishment of cAMP-dependent regulation in HCN2EA mice generates thalamocortical neurons that cannot switch from burst to tonic firing in response to sensory imputes or with an increase in intracellular cAMP concentrations [23]. On the behavioral level, the CNBD site mutation makes the thalamic network unable to maintain transmission mode during wakefulness, which leads to recurrent absent epilepsy and behavioral arrest [23]. Therefore, we infer that the xenon-induced inhibition of the HCN2 channels via the CNBD site might affect the thalamocortical network function in transmission mode and facilitate a switch to burst firing that suppresses thalamocortical signal propagation. This is consistent with previous studies showing the consequences of inhibition of I_*h*_ currents by different anesthetics [44]. Overall, these inferences highlight the complex mechanisms of xenon inhibition of HCN2 channels, its effect on thalamocortical network regulation, and sensory information transfer to the cortex.

Xenon’s inability to inhibit HCN2 channels in HCN2EA mice was mirrored by a reduction in the xenon sedative effect when applied at (70%) in the open-field test. This concentration represents, based on mice MAC values, a sub-anesthetic concentration that is sufficient to induce sedation [45]. The mice’s behavior in the open-field test is exploratory in the first 5 min and adaptive after 15 min. In the open field test, wild-type mice and HCN2EA mice showed similar behavior during the habitation and exploration period and with the addition of control gas. When the latter was replaced with the xenon mixture, the anesthetic did not produce a full sedative effect on HCN2EA mice after 10 min of exposure, unlike their wild-type counterparts.

Mutating the CNBD site of HCN2EA had no discernable effect on HCN2 channel expression and distribution [23] in the HCN2EA mice brain. Moreover, HCN2EA mice spent an equal amount of time in wakefulness as their wild-type counterparts [23] owing to the presence of several cAMP-independent signaling pathways that ensure a relatively normal, in terms of time spent in each conscious state, sleep-awake cycle [46,47]. Therefore the observed in vivo results clearly demonstrate that the lack of a complete sedative effect in HCN2EA mice is the result of the mutated CNBD form on HCN2 channels and not a change in the HCN2EA mice physiology.

Interestingly, HCN2EA mice were not completely unaffected by xenon application and nevertheless showed a decreased activity level of around 30%. This observation leaves open additional mechanisms through which xenon produces anesthesia. As with most anesthetics, xenon interacts with different targets in the CNS by binding to macromolecule cavities and altering their biological functions [48] to mediate anesthetic and analgesic effects [49]. Xenon antagonizes AMPA receptors and NMDA receptors with low potency [50], NMDA receptors presumably by binding to the NMDA-glycine binding site [51]. Xenon can also enhance the activity of other receptors that influence neuron activities such as the TREK channel [52] and the ATP-sensitive potassium channel [53]. Furthermore, it should be noted that HCN2EA mice still express functional HCN4 channels that might be modulated by xenon since they are expressed by thalamocortical neurons and have a similar CNBD site [21,54]. We hypothesize that the pharmacological activity against those targets and the insensitivity of HCN2 channels in the HCNEA mice toward xenon explain why the mice activity decreases without reaching wild-type sedation levels. This leads to the conclusion that a part of the xenon hypnotic effect is dependent on HCN2 channel inhibition by interference with the channel CNBD.

## 4. Materials and Methods

### 4.1. Acute Thalamocortical Brain Slice Preparation

Animal handling was approved by the competent veterinary office (Munich, Germany). We used C57Bl/6N and HCN2EA mice of both gender (P28-P35) for acute thalamocortical slice preparation. Briefly, mice were deeply anesthetized with isoflurane before head decapitation. Mice brains were quickly extracted and transferred for cutting on a vibratome (HM 650V, Microme) in an ice-cold sucrose cutting solution continuously oxygenated by carbogen (95% O_2_, 5% CO_2_) and optimized for slice preparation (mM): 2.5 KCl, 1.25 NaH_2_PO_4_, 26 NaHCO_3_, 0.5 CaCl_2_, 7 MgCl_2_, 105 sucrose, 24.7 glucose, and 1.7 ascorbic acid. Horizontal slices containing the VB region of interest (3–6 slices) were made at 250 µM thickness and transferred for storage to a carbogen-saturated artificial cerebrospinal fluid solution (aCSF) made of (mM): 131 NaCl, 2.5 KCl, 1.25 NaH_2_PO_4_, 26 NaHCO_3_, 2 CaCl_2_, 1.2 MgCl_2_, 18 glucose, and 1.7 ascorbic acid at 34 °C in a warm-bath for 30 min followed by another 30 min at room temperature.

### 4.2. Whole-Cell Voltage Patch Clamp Recordings

For whole-cell patch-clamp recordings, individual slices were transferred to the recording chamber and mechanically fixated with a standard grid. The chamber was continuously perfused with a carbogen-saturated aCSF solution topped with 1 mM BaCl_2_ and 0.5 µM Tetrodotoxin TTX Tocris Bioscience (Wiesbaden-Nordenstadt, Germany), at an adjusted temperature of 31 °C to 33 °C. Ba_2+_ is routinely used in I_*h*_ current recording to inhibit the potentially interfering effects on inwardly rectifying potassium channels (Kir) and members of the two-pore domain acid-sensitive potassium channels (K_2_P), as well as facilitate improved analysis by significantly enhancing the voltage sag amplitude [55].

Upon visual allocation of the thalamic VB (Axioskop 2FS plus, Zeiss Zeiss, Oberkochen, Germany), thalamocortical neurons somata were identified using infrared videomicroscopy (ORCA-R2, hamamatsu (Herrsching, Germany). Patch pipettes were made from borosilicate glass tubing (GC150TF-10, Harvard Apparatus Massachusetts, USA) using a horizontal puller (MDZ Universal Electrode puller, Zeitz-Instruments Munich, Germany). The pipettes were fire polished and filled with an intracellular solution containing (mM): 140 KMeSO_4_, 10 HEPES, 10 KCl, 0.1 EGTA, 10 phosphocreatine, 4 MgATP, and 0.2 GTP. The osmolarity was adjusted to 305 mOsm/l, and the pH was adjusted to 7.3. For experiments with 1 µM cAMP, 8-bromo cAMP (Sigma-Aldrich, B7880) was added to the pipette solution. The final resistance of the electrode was between [2.5–6.5] MΩ. All chemicals were brought from Sigma-Aldrich (Steinheim, Germany).

For I_*h*_ measurement, thalamocortical neurons in the whole-cell configuration were clamped using a HEKA amplifier EPC 10 USB Double Patch Clamp Amplifier, HEKA (Stuttgart, Germany) at −40 mV from which a pulse of 4 s duration was applied from −140 mV to −40 mV in 10 mV increments (Figure 1A, pulse protocol) followed by a 1 s test pulse to −140 mV. Respective tail current amplitude I_*tail*_ was normalized to the tail current acquired at the −140 mV voltage step. The obtained data were then normalized in graph prism (GraphPad, San Diego, CA, USA) using the normalization function where 100% represented the highest value and 0% the lowest, which ensured that all curves would decay to zero. The curves were then plotted and fitted with the Boltzmann equation with constrained top and bottom plateaus to the constant values of 1 and zero, respectively. Outliners were not eliminated but had a limited effect on the regression model (robust regression). The Boltzmann equation $I/I_{max}=\left(A_{1}-A_{2}\right)/\left(1+e^{[V-V_{1/2}]/k}\right)+A_{2}$ was applied to estimate steady-state activation and calculate the half-maximal activation of I_*h*_, ($V_{1/2}$). In this equation, $V_{1/2}$ and k represent the half-maximum activation and the Boltzmann slope factor while *A*_1_ and *A*_2_ represent initial and final I/I_*max*_ values, respectively. The current response was amplified, low-pass filtered at 3 kHz, and digitized at 20 kHz. Fast and slow capacitive transients were canceled by the compensation circuit of the EPC-10 double amplifier. Data from individual cells were discarded if they showed no GΩ-seal resistance formation upon establishment of cell-attached configuration, resting membrane potential below −50 mV, unstable series resistance, or holding current (>2% change).

### 4.3. Application of Nitrogen and Xenon Gas Mixture

All our recordings were performed in a continuously oxygenated aCSF recording chamber. Carbogen-alone recordings were performed on the first thalamocortical neuron that was patched in a fresh slice before any additional gas mixture was added to the aCSF. Xenon gas mixture used in whole-cell, patch-clamp experiments was composed of (65% xenon, 30% O_2_, 5% CO_2_) and was co-applied with the carbogen mixture in the aCSF (Figure 2A, schematics) using two microfilter candles (Bioenno tech Lifesciences, Santa Ana, CA, USA; one for each gas. Xenon I_*h*_ recordings were taken after a minimum of 30 min of xenon gas mixture co-application with carbogen. This allowed full equilibration of the xenon mixture with the recording chamber and achieved a concentration of 1.90.2 mm [50]. A nitrogen gas mixture composed of (65% N_2_, 30% O_2_, and 5% CO_2_) was bubbled with carbogen (Figure 2A, black schematic) in a separate set of experiments performed on different brain slices to obtain control recordings. Control recording had the same minimum waiting time as its xenon counterpart. We recorded between 1 and 3 thalamocortical neurons from the same slice if it was exposed to one of the additional gas mixtures. All gas mixtures were purchased from Linde AG (Unterschleissheim, Germany).

### 4.4. Modified Open-Field Test

Experiments with live animals were covered under the license **ROB-55.2-2532.Vet_0212-111**. To test the effect of xenon sedation in vivo, an adopted open-field test was used as reported previously [11]. Briefly, mice were acclimated to the experimental room overnight. The animal to be tested was placed in a Custom madebox (40 × 20 × 20 cm), which was connected to two gas cylinders via a three-way valve either containing the control gas mixture (70% N_2_, 30% O_2_) or xenon gas mixture (70% xenon, 30% O_2_) (Figure 4A,B). The bottom of the experimental box was gridded, and the behavior was recorded on Appendix A. Animals were adopted 5 min to the open field environment. Additionally, at a time point of 15 min, the open field chamber was perfused with the control gas mixture at a flow rate of 0.5 L/min in order for the animals to become used to the gas perturbation. After this initial phase of adaptation, the xenon gas mixture was injected into the chamber, and spontaneous activity was recorded for an additional 10 min. After the open field test, the mouse was returned to its home cage. For every animal, the ratio of lines crossed was analyzed. The xenon-mediated effect was calculated as the ratio of lines crossed after xenon treatment to the lines crossed under basal conditions (70% N_2_, 30% O_2_). All gas mixtures were purchased from Linde AG (Unterschleissheim, Germany).

### 4.5. Statistical Analysis

For statistical analysis and the creation of the different graphs, GraphPad Prism 9, San Diego, CA, USA was used. For plotting representative I_*h*_ traces, Matlab was used. I_*h*_ recordings were imported into Matlab with the HEKA import function [56] then rescaled to values between 0 and 1 and normalization before plotting. I_*h*_ currents were analyzed using FitMaster (HEKA Elektronik, Harvard Bioscience Holliston, MA, USA). The sample sizes for different experiments were chosen based on previous preliminary data. In our ex-vivo the I_*h*_ activation curve values were plotted as mean+SEM in the function of membrane voltage from (−140 mV to −40 mV) with the Boltzmann fitting on top. Since our data sample was not large enough for good interpretation by normal distribution testing [57], we opted for the application of non-parametrical testing by default. Accordingly, all comparisons of half-maximum activation that were plotted with statistical testing are represented as medians with the interquartile range (IQR). For multiple comparisons, we used the Kruskal–Wallis test with a post hoc false discovery rate correction (The Benjamani, Krieger, and Yekutieli method). We reported the uncorrected *p*-values in the text for a single comparison within the Kruskal–Wallis test and the corrected one reporting *p*-value for the whole test. Only plotted comparisons were included in the analysis. We also reported the effect size of our comparisons as an area under the receiver operating characteristic curve (AUC) values with their respective confidence intervals. The effect size was considered large enough if the AUC was above 0.7 and the respective confidence interval did not contain the 0.5 value [58]. For the in-vivo study, numerical data are presented as the mean+SEM with the number of experiments, (cell recordred) indicated, if not stated otherwise. If error bars are not visible in graphs, the bars are smaller than the symbol size. All statistical tests were performed on a two-sided level of significance of 5% and were indicated by asterisks in the figures.

## 5. Conclusions

In summary, our findings suggest that xenon’s sedative properties are partly attributed to interfering with the CNBD of HCN2 channels. As a result, the activity of these channels is reduced in thalamocortical neurons, leading to a decreased activity of neuronal transmission to the cortex. Our experiments conducted both ex vivo and in vivo using the HCN2EA mouse model, indicate that modifying the binding site reduces the effects of xenon on cellular and behavioral levels. Overall, these findings shed light on the complex molecular mechanisms underlying the anesthetic properties of xenon.

## Figures and Tables

**Figure 1 ijms-24-08613-f001:**
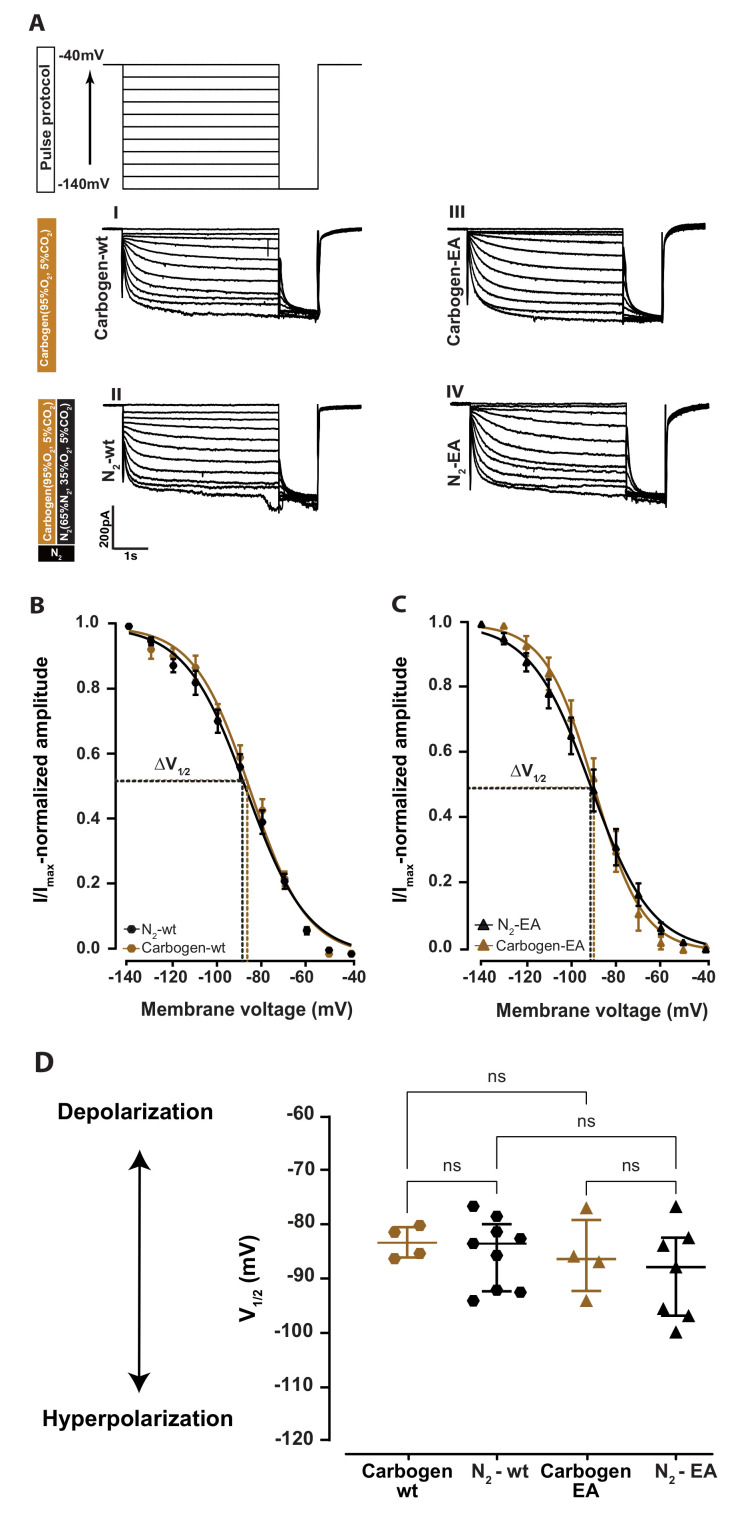
Co–application of nitrogen gas mixture with carbogen gas mixture in the aCSF does not affect I_*h*_ and the $V_{1/2}$. (**A**) The pulse protocol that was used to generate every HCN2 currents I_*h*_. I_*h*_ current were generated as described by applying a command voltage in 10 mV increments from −140 mV to −40 mV. Schematic representation of the applied gas mixture (nitrogen: black, carbogen: brown) with the percentages of the different components in each. Representative I_*h*_ current traces from thalamocortical neurons during carbogen alone (wild-type: I, HCN2EA: III) with the addition of nitrogen (wild-type: II and HCN2EA: IV). In thalamocortical neurons of wild-type (**B**), HCN2EA (**C**), the application of nitrogen gas mixture to the aCSF did not change the I_*h*_ current activation curve to a different potential than the carbogen-alone application, the curve fitting is used to calculate the half-maximum voltage activation $V_{1/2}$ as shown by the dotted lines. (**D**) Nitrogen co-application with the carbogen gas mixture did not result in significant changes of $V_{1/2}$ compared between the different groups. $V_{1/2}$: carbogen-wt: −85.52 [−88.22–−82.65] mV N_2_-wt: −85.67 [−94.74–−82.10] mV carbogen-EA: −90.03 [−98.99–−84.59] mV N_2_-EA: −88.52 [−94.42–−81.30] mV. *p* = 0.5661; Kruskal–Wallis test with post hoc false discovery rate correction (FDR), four different comparisons. ns: no significant results.

**Figure 2 ijms-24-08613-f002:**
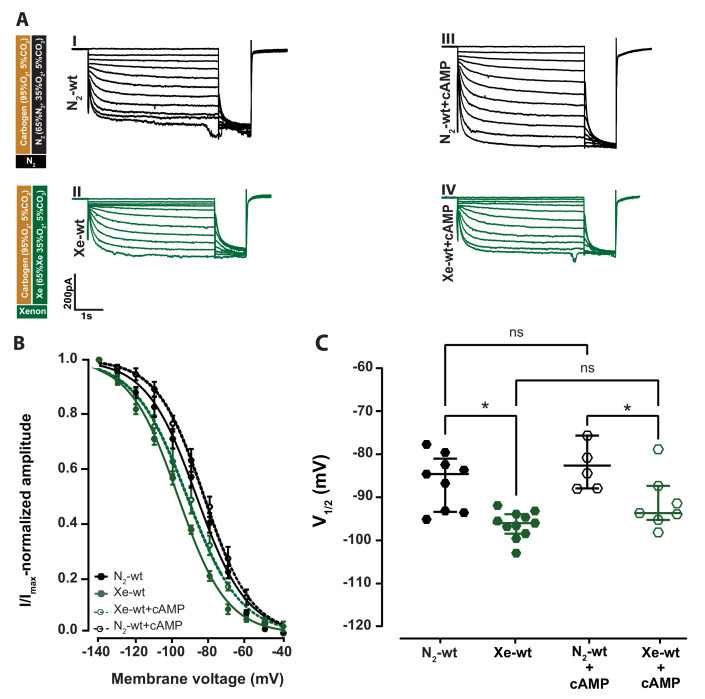
Xenon shifts the I_*h*_ activation curve of the HCN2 channel toward more hyperpolarized potential independent of 1 µM cAMP. (**A**) Schematic representation of the different gas applied with the proportions of the different components in each. Representative II_*h*_ curve’s from the thalamocortical neurons of wild-type mice during control N_2_ and xenon application without cAMP (I and II) and with 1 µM cAMP (III and IV), respectively. (**B**) Xenon application to thalamocortical neuron with (open green hexagons) and without (filled hexagons) 1 µM cAMP leads to a significant shift toward more hyperpolarized potential compared to respective controls (open black and filled black hexagons, respectively). (**C**) Xenon application for 30 min resulted in an shift of $V_{1/2}$ to more hyperpolarization compared to controls $V_{1/2}$ N_2_-wt = −85.67 [−94.47–−82.10] mV, Xe-wt = −97.09 [−99.56–−95.04] mV, *p* = 0.0009; N_2_-wt+cAMP = −83.69 [−89.03–−76.72] mV, Xe-wt+cAMP = −94.78 [−96.36–−88.44] mV, *p* = 0.046; Kruskal–Wallis test. Significant results are indicated by asterisks. No statistically significant results were found between the $V_{1/2}$ of N_2_ and N_2_-wt+cAMP, *p* = 0.42 or the Xe and Xe-wt+cAMP, *p* = 0.11; Kruskal–Wallis test. ns: no significant results. significant result at indicated by asterisks.

**Figure 3 ijms-24-08613-f003:**
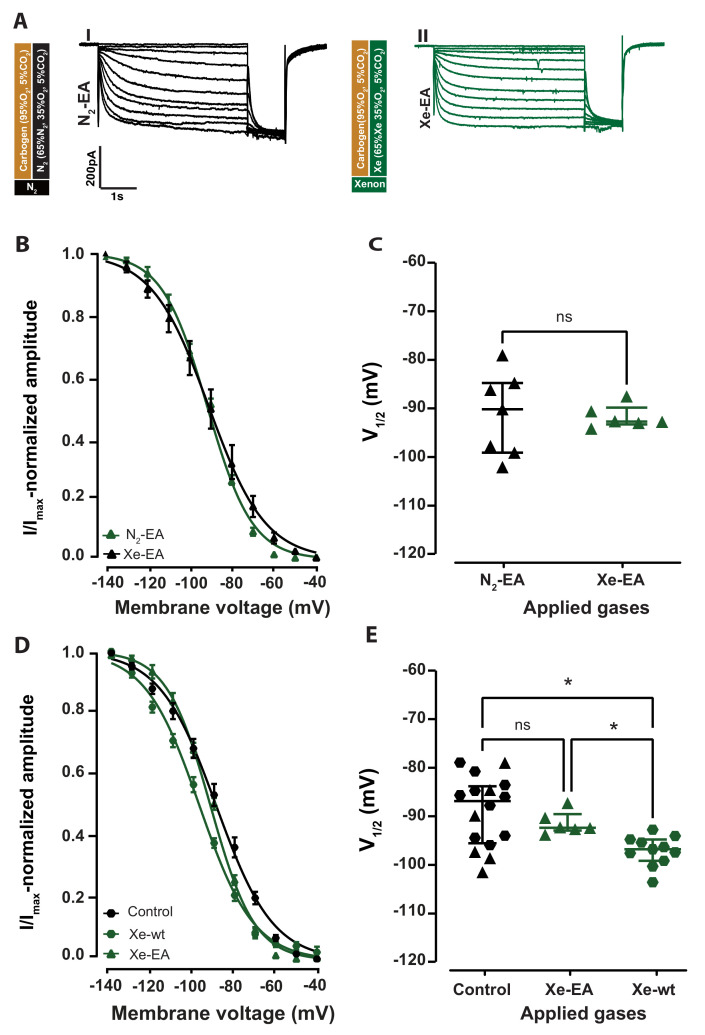
Mutation of the cyclic nucleotide-binding domain (CNBD) of HCN2 channels abolishes xenon effect. (**A**) Representative I_*h*_ curves from the thalamocortical neurons of HCN2EA mice during N_2_ and xenon application (I, II, respectively). (**B**) Xenon application for more than 30 min did not shift the I_*h*_ current activation curve (green triangles) toward more hyperpolarized potentials compared to the control (black triangles). (**C**) The calculated $V_{1/2}$ from the fitted traces show no significant reduction in $V_{1/2}$ after xenon application for 30 min −92.56 [−93.16–−89.68] mV compared to control −90.03 [−98.99–−84.59] mV, *p* = 0.84; two-tailed Mann–Whitney test. (**D**) Xenon significantly shifts the activation curve I_*h*_ of wild-type neurons (green hexagons), but not of HCN2EA (green triangles) compared to the Control group (black hexagons). (**E**) Xenon application resulted in $V_{1/2}$ hyperpolarization in wild-type Xe-wt $V_{1/2}$: −97.09 [−99.56–−95.04] mV, green hexagons, p_*wt*_ = 0.0012, but not HCN2EA Xe-EA $V_{1/2}$: −92.56 [−93.16–−89.68] mV, green triangles, p_*EA*_ = 0.91 compared to controls control $V_{1/2}$: −86.94 [−95.85–−83.79] mV, black hexagon; Kruskal–Wallis test. A significant difference was found between Xe-wt and Xe-EA $V_{1/2}$
*p* = 0.01; Kruskal–Wallis test. Three different comparisons. Significant results are indicated by asterisks. ns: no significant results.

**Figure 4 ijms-24-08613-f004:**
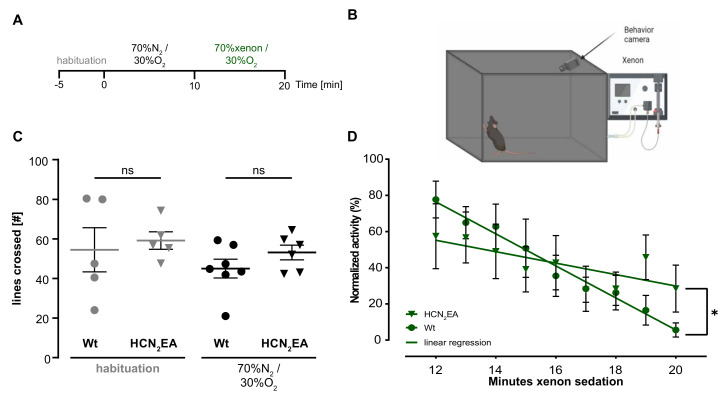
In-vivo xenon sedation is significantly reduced in HCN2EA mice. (**A**) Schematic of the adopted open field timeline to test the sedation effect of xenon. (**B**) Experimental setup for the test. (**C**) Analysis of the habituation (gray) and adaptation with control gas (70% N_2_, 30% O_2_) (black) revealed no differences in the number of crossed lines in wild-type (circles) to HCN2EA animals (triangles). So, neither the open field box itself (habituation) nor the noise of the gas influx showed genotype differences. $p_{\left(habituation\right)}$ = 0.7069; *p*_(*control*)_ = 0.2128; two-tailed Mann–Whitney test. (**D**) The linear fitting of the normalized activity shows a drop over a 10 min xenon sedation period (min 10 to 20 of the experiment, green) in wild-type animals (circles), but not in HCN2EA animals (triangles). $p_{\left(xenon\right)}$ = 0.0006, 20.38% variation; two-way ANOVA. Significant results indicated by asterisks. ns: no significant results.

## Data Availability

The raw data supporting the conclusions of this article will be made available by the corresponding author upon reasonable request, without undue reservation.

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
