# Peer review of "Xenon’s Sedative Effect Is Mediated by Interaction with the Cyclic Nucleotide-Binding Domain (CNBD) of HCN2 Channels Expressed by Thalamocortical Neurons of the Ventrobasal Nucleus in Mice"

_ijms, 2023, doi:10.3390/ijms24108613_

Round 1

Reviewer 1 Report

The authors here investigated the role of the cyclic nucleotide-binding domain of HCN2 channels in the Xenon’s sedative effect on HCN2 channels. This was done by using the transgenic mice model HCN2A with two mutations in the cyclic nucleotide-binding domain to remove the binding of cAMP to HCN2 channels. They found that xenon shifted the voltage dependence of current activation to more negative voltages in thalamocortical neurons in the wt but not the HCN2A model. In the open field test, the xenon application reduced mice activity in the wt but not the HCN2A model. The authors concluded that Xenon inhibits HCN2 channels by mediating with the cyclic nucleotide-binding domain. The data seem convincing, and the conclusion seems appropriate. However, I have some concerns below.  

1. Is it possible to add the voltage pulse protocol in Figure 1?

2. Figure 1D, the y axis is V1/2 not deltaV1/2.

3. Line 129, as the authors mentioned in reference 11, has the authors tried applying 30 uM to fully abolish Xenon effects in wt mice?

4. Line 144, I don’t understand why the authors combined the wt control and EA control together as one control group.

5. Grammar errors, such as lines 299 and 72.

None. 

Author Response

We thank the first reviewer for his kind and insightful comment on our manuscript. We appreciate his suggestion that can definitely improve the manuscript further and allow an easier understanding. We would like to provide the following clarification and answers.

  • Is it possible to add the voltage pulse protocol in Figure 1?
    • As requested by the reviewer a protocol pulse was added to Figure 1 between the representative Ih traces with the proper voltage indication and a description in the caption.
    •  In Figure 1D, the y-axis is V1/2 not deltaV1/2.Thank you for pointing this mistake out. We never intended to point to a difference between the V1/2 values and therefore we crossed the delta from all the graphs that show a V1/2 in figure1, 2, and 3.
    •  Line 129, as the authors mentioned in reference 11, has the authors tried applying 30 µM to fully abolish Xenon effects in wt mice? The authors of this manuscript have not done experiments with 30 µM cAMP intracellular solution. This concentration was however previously investigated and reported by Mattusch et al, 2015. The authors of that paper reported that 30 µM cAMP is a saturating concentration capable of blocking the effects of xenon on HCN2 channels in experimental conditions identical to the one in this manuscript (ex-vivo patch-clamp).

  •  Line 144, I don’t understand why the authors combined the Wild-type control and EA control as one control group
    • The decision to pool the different control groups into a single one was based on 3 assumptions:
      1. The authors found no statistical differences between the two-control group N2 wild-type and N2-EA in terms of statically testing, effect size (AUC), and the confidence interval of the AUC as reported in the manuscript. The values also overlapped and did not form any visible clusters. This suggested that the groups were sufficiently similar to be combined.
      2. The sample size of the wild-type and HCN2EA group is unequal and might be a source of bias
      3. Combining the control group data can mitigate this risk and increase the statistical power of the comparison, providing a more robust analysis.
  • Grammar errors, such as lines 299 and 72
    • We thank the reviewer for taking the time to point out some of the grammar mistakes and typos in our manuscripts. The manuscript was revised again and the mistakes in lines 299 and 72 were corrected along with other similar mistakes.

Reviewer 2 Report

In this paper, the authors demonstrated that the sedative effect of the gaseous anesthetic xenon was strongly dependent on the CNBD site of HCN2 channels. Using whole cell patch clamp recording, they found that xenon caused a significant leftward shift in the V1/2 of Ih in wild-type mice; this effect was significantly diminished in thalamocortical neurons of HCN2EA mice. they also applied the open-field test and showed that xenon interference with the CNBD was necessary for mediating part of xenon’s sedative effect. Actually, previous studies have already demonstrated the effects of xenon on HCN2 channels. And this paper paid more attention to the interaction with the cyclic nucleotide-binding domain of HCN2channes. This can be interesting to the Xenon’s sedative effect, but further studies are needed to support their findings.

1.       The ex-vivo patch-clamp recording results are good. Is there any in vivo electrophysiological data to confirm and strength the effect of xenons on HCN2 channels?

2.       To test the xenon-mediated hypnotic properties, more behavior tests are needed in addition to the open-field test.

3.       Line 72, in order to investigate the effect of xenon interacting with the CNBD of HCN2 channels on the electrophysiological properties of the channel……, “in” should be capitalized.

4.       Line 171, The difference was not statistically significant on the……, The sentence is incomplete.

The language is good and understandable.

Author Response

We thank the second reviewer for his insightful comment on our manuscript. We would like to clarify the following points and issues raised by the reviewer.

  • The ex-vivo patch-clamp recording results are good. Is there any in vivo electrophysiological data to confirm and strengthen the effect of xenon on HCN2 channels?
    • Although we do agree in principle that an in-vivo electrophysiology is in theory feasible we would like to point out the following:
      1.  An in-vivo calcium imaging experiment to measure thalamocortical neuron activity during xenon application would require an invasive craniectomy, likely causing disturbances to experimental readout caused by stress, different anesthesia exposure, and potential pain. Moreover, this will not give a direct readout of HCN2 channels and the interaction with cAMP, rather it will focus on the neuron’s behavior and firing pattern during xenon application. 
      2. An in-vivo patch-clamp If successful the readout will also suffer from the same problems described above and the experimental results can not be considered pure HCN2 since it’s not possible to manipulate the extracellular environment.
      • In both in-vivo setups, an anesthetic adjuvant must be added such as dexmedetomidine since it’s not possible to completely sedate mice with xenon alone as stated in the manuscript which will contaminate a pure xenon effect. For those reasons, we believe that the ex-vivo recording reported in our manuscript is sufficient to support the reported conclusions
  1. To test the xenon-mediated hypnotic properties, more behavior tests are needed in addition to the open-field test.
    • We agree with the reviewer’s comment that one single open-field test does not test all hypnotic properties of xenon. We also agree that a sedative effect might also change concentration or learning which at first is a completely independent (mostly hippocampal) pathway. In this paper, we solely wanted to measure the sedative effects caused by the application of xenon which we defined as a reduction in movement. To abolish movement completely by sedating a rodent with xenon is not possible due to the MAC value limitation in the experimental setup. It is our opinion that additional behavioral experiments won’t add additional value to our study because the analysis of motor behavior is the most important parameter.
  1. Line 72, in order to investigate the effect of xenon interacting with the CNBD of HCN2 channels on the electrophysiological properties of the channel……, “in” should be capitalized
    • We thank the reviewer for taking the time to point out some of the grammar mistakes and typos in our manuscripts. The manuscript was revised again and the mistakes in lines 299 and 72 were corrected along with other similar mistakes.
  1. Line 171, The difference was not statistically significant on the……, The sentence is incomplete.
    • Thanks for pointing this out. That’s a typing error, the sentence should be, ‘The difference was not statistically significant (Figure…)'. We changed it in the text.